# Evolution of Medical Approaches and Prominent Therapies in Breast Cancer

**DOI:** 10.3390/cancers14102450

**Published:** 2022-05-16

**Authors:** Suzann Duan, Iain L. O. Buxton

**Affiliations:** 1Department of Pharmacology, University of Nevada, Reno School of Medicine, Reno, NV 89557, USA; sduan@email.arizona.edu; 2Department of Medicine, University of Arizona College of Medicine, Tucson, AZ 85724, USA

**Keywords:** breast cancer, hormonal therapy, immunotherapy, seed and soil hypothesis

## Abstract

**Simple Summary:**

Breast cancer is a leading cause of morbidity and mortality in women worldwide, with mortality rates largely driven by metastatic disease. Despite concerted efforts to uncover and target the mechanisms underlying these events, the five-year relative survival rate for metastatic breast cancer remains an abysmal 27%. To better inform future directions for managing this disease, it is helpful to examine the evolution of cancer research, from foundational concepts to therapies and their pitfalls. This review aims to provide a rich overview of the history of breast cancer therapies spanning antiquity to the modern era, concluding with the rapid advancements in immunotherapies and our eventual progression towards personalized medicine.

**Abstract:**

An examination of the origins of medical approaches to breast cancer marks this disease as one of the most difficult to manage. As the early identification, diagnosis and treatment of breast cancer evolve, we will move to a time when each patient and their cancer can be assessed to determine unique patient-specific (personalized) approaches to therapy. Humans have attempted to manage breast cancer for millennia. Even today, the disease claims thousands of lives each year. In light of the increasingly sophisticated understanding of cancer diagnosis and treatment, together with our ultimate failure to offer a cure in the most difficult cases, it is instructive to reflect on the beginnings of our understanding.

## 1. Introduction

### 1.1. Origins in Antiquity

Long regarded as a symbol of allurement and fertility, few aspects of the human physique are subject to greater praise and scrutiny than the female breast. Under the sharpened gaze of a physician-scientist, however, the breast renders a unique quandary dating back to the 17th century B.C.E. Perplexed by incidences of a disfiguring and incurable malady of the breast, ancient Egyptian physicians were among the first to note the organ with medical interest. Some experts contend that these first accounts of breast cancer were recorded in the *Edwin Smith Surgical Papyrus* by the infamous Egyptian physician Imhotep himself [1]. The ancient text was acquired in 1862 by the American collector Edwin Smith and documents the first medical rationale for treating tumors of the breast with cauterization [2]. The Egyptian treatise remains one of four medical papyri dating back to antiquity and demonstrates a profound grasp of surgery and medicine that rivaled the practice of Grecian physicians born a millennium later. 

Ancient Hellenistic writings further documented early cases of breast cancer and introduced the usage of modern medical terminology to describe carcinoma and metastatic disease [3]. The word cancer itself lends its origins to the Greek physician Hippocrates (c. 460–370 B.C.E.), who likened a tumor and its vascular fragments to the depiction of a crab [4]. Hippocrates postulated that imbalances in four bodily humors preceded disease, including tumors, and treatment during this period largely focused on dietary intervention and the then-common practice of bloodletting. Such minimally invasive treatments were aligned with ancient Greek tradition that frowned upon surgical intervention [5]. While his humor-centric theory was later disapproved, Hippocrates’ early distinction between benign and malignant disease remains a pillar of modern oncology. Strongly influenced by Hippocrates’ teachings, the Greek physician and philosopher Claudius Galen (c. 129–216 C.E.) further contributed to the *carcinos* nomenclature first introduced by his predecessor. The modern derivation of the term oncology originated from Galen’s usage of the Greek word *oncos*, meaning swollen, to describe all tumors [4]. Galen’s belief that cancer manifested from an accumulation of black bile, a humor previously born from Hippocratic theory, remained a prevailing doctrine for centuries to come [6]. 

Medical practice in the subsequent millennium (c. 500–1500 C.E.) was dominated by the dissemination of religious philosophies that favored faith-based healing over the seemingly barbaric nature of invasive surgery. During a period marked by surgical stagnation, cauterization and application of arsenic-containing caustic pastes were preferred treatments for tumors of the breast [7]. The invention of the compound microscope by Dutch spectacle-maker Zacharias Janssen in 1595 ushered a new era of medicine characterized by the emergence of surgery during the 18th and 19th centuries [8]. Refinement of the single-lens microscope by Dutch scientist Antony van Leeuwenhoek (1632–1723) and his subsequent identification of bacteria and blood cells became an inflection point in medical history. Van Leeuwenhoek’s discoveries, combined with those made by Robert Hooke and other prominent microscope enthusiasts of the time [9], marked a departure from the metaphysical teachings of Hippocrates and Galen in favor of the empiricism of cellular pathology [10]. Such revelations were critical to catalyzing the golden age of discovery and catapulted Europe into a new era of cancer research and pathology.

### 1.2. A Cellular Basis for Cancer

Cases of lumpectomies, mastectomies, and axilla removal began to fill European medical records, having been touted by prominent surgeons such as Jean Louis Petit (1674–1750) and Lorenz Heister (1683–1758) [11]. The 19th-century surgical practices saw growing success rates with the introduction of anesthesia (1846) and antiseptic techniques (1867) [12,13]. Such advancements led to a definitive rejection of Hippocrates’ humor-centric view in favor of lymphatic theory and exploration. French physician Henri Le Dran (1685–1770) first noted that breast cancer could infiltrate the axillary lymph nodes, resulting in a poorer prognosis [11]. Similarly, Scottish surgeon John Hunter (1728–1793) cited the lymphatic system as a direct source of cancer, from which it could invade into other tissues [11]. 

The theory that lymph and cancer were synonymous entities greatly weakened after German pathologists Johannes Muller (1801–1858) and Rudolf Virchow (1821–1902) detailed a cellular basis for tumors [14]. A prolific writer and scientist, Virchow released several works including his acclaimed *Die Cellularpathologie* (1858), describing the cellular origins of cancer [15]. Virchow postulated that cancer cells derived from pre-existing cells through a process stimulated by chronic inflammation. Prominent English pathologist and surgeon James Paget (1814–1899) further contributed to the pathological basis of cancer through his *Lectures on Tumours* (1851) and *Lectures on Surgical Pathology* (1853) [16,17]. Paget observed physical changes to the nipple and areola of the breast 1–2 years prior to tumor onset, leading to the first description of Paget’s disease as detailed in his 1874 publication *On Disease of the Mammary Areola Preceding Cancer of the Mammary Gland* [18]. As a result of their collective contributions, Virchow and Paget are frequently cited as the founding fathers of cellular pathology [19,20].

A comprehensive understanding of how cancers spread remained a lacking feature of the early 19th century. In his 1829 publication *Recherches sur le Traitement du Cancer* [21], French gynecologist Joseph Recamier (1774–1852) coined the term metastasis to describe the spread of cancer to blood and other tissues [22]. Karl Thiersch (1822–1895) further refined this concept by demonstrating that metastasis arose from the spread of malignant cells rather than from blood itself [11]. Not to be usurped by his father’s legacy, Stephen Paget (1855–1926) made the critical observation that the distribution of malignant growths among secondary organs was non-stochastic. Paget examined the necropsy results of 735 fatal cases of breast cancer and found that secondary growths manifested more frequently in the liver (241 cases or 33%), lungs (70 cases or 10%), ovaries (37 cases or 5%), and bone (at least 14 overt cases with additional reports of bones afflicted by brittleness) [23]. Paget concluded in his famed *The Distribution of Secondary Growths in Cancer of the Breast* (1889) that metastatic tumor cells, despite being ubiquitous in the bloodstream, only grew at compatible tissue sites. Similar to a plant’s seeds being disseminated by the wind, Paget argued that malignant cells carried by blood preferentially survived and grew through elected interaction with the “congenial soil” of remote organs [23]. Having stood the test of time, Paget’s “seed and soil” hypothesis remains steadfast in shaping modern-day views of metastasis.

## 2. The Reign of the Knife

Expanding knowledge on cancer’s cellular origins heralded a blossoming awareness for more sophisticated treatment regimens. As a result, the late 19th and early 20th centuries saw major advancements in both surgical and alternative therapies for breast cancer [11]. In 1894, American surgeon William Halsted developed the radical mastectomy, arguing for the precise and complete removal of the afflicted breast, axillary lymph nodes, and *pectoralis major* to limit further recurrence [24,25,26]. Halsted’s procedure was quickly adopted as the leading standard of care for breast cancer treatment until modifications to preserve the chest wall and axilla were introduced several decades later [27]. Popularity in radical mastectomies rapidly declined during the second half of the 20th century as cancer became increasingly viewed as a systemic disease [28,29]. A shift toward less invasive procedures coincided with the results of a large-scale prospective clinical trial comparing radical mastectomy with alternative treatments. The study, which detailed a postoperative follow-up of 1665 patients (2–6 years post-treatment), showed no added benefit of radical mastectomies compared to more minimally invasive procedures [30]. Surgical treatments circumventing the breast also began to emerge during Halsted’s reign. Sir George Thomas Beatson (1848–1933) discovered that bilateral oophorectomy resulted in improved outcomes for patients with advanced breast cancer. Published in his 1896 *On Treatment of Inoperable Cases of Carcinoma of the Mamma*, Beatson’s findings were among the first to suggest anti-hormonal treatment for advanced breast cancer [31]. 

### 2.1. The Race toward Radiotherapy

A series of seminal discoveries made at the cusp of the 20th century birthed a new paradigm for cancer treatment and revolutionized modern oncology. Following the discovery of X-rays in 1895 by Wilhem Roentgen (1845–1923), scientists across the globe raced to understand and harness the properties of radiation. Emerging at the forefront were French physicists Henri Becquerel (1852–1908) and Pierre Curie (1859–1906), and Polish-French physicist and chemist Marie Sklodowska Curie (1867–1923) [32]. The trio was awarded the 1903 Nobel Prize in Physics for their revolutionary work on uncovering natural radioactivity [33]. Prior to their discoveries, radiation, though previously shown to kill diseased cells, remained poorly understood and its use in medicine lingered in its infancy. In 1896, Victor Despeignes (1866–1937) and Emil Grubbe (1875–1960) were among the first to use X-ray machines on cancer patients (diagnosed with cancer of the stomach and breast, respectively), but were met with limited success [34,35]. As knowledge on radioactive elements and methods for their isolation grew, radiotherapy quickly emerged as a powerful tool to combat malignancy. 

Pierre and Marie Curie’s discovery of radium and polonium, which earned the latter the 1911 Nobel Prize in Chemistry, further contributed to the use of radiotherapy to combat cancer. In contrast to the X-ray, which could only be administered over a broad surface, radium could be delivered locally, topically (via radium salts), and interstitially (via radium emanating rods) [36]. Thus, radium therapy prevailed as a popular treatment for solid and cutaneous malignancies [37,38,39,40,41], while the X-ray remained a pragmatic approach for treating lymph-infiltrating cancers [42,43]. Marie Curie’s remarkable achievements also had far-reaching implications in cancer imaging and diagnostics; her isolation and application of radioisotopes enabled future studies on the use of radiolabeling for intravital imaging of solid tumors [44]. Knowledge on the detrimental health effects of radium exposure surfaced during the latter half of the 20th century, leading to its succinct demise as a monolith of radiotherapy. Subsequent precautions and modifications in X-ray and radiation technologies have ensured their continued use throughout modern history. Nevertheless, plateauing remission rates and safety concerns associated with radiotherapy signaled the emergence of a second pillar of oncology: chemotherapy. 

### 2.2. The Second Coming: Chemotherapy

The conception of chemotherapy can be ascribed to the prominent chemist Paul Ehrlich (1854–1915), whose research aimed to discover a “*therapia sterilisans magna*” (great sterilizing therapy) for the treatment of bacterial disease [45]. With its modern birthplace entrenched amidst the carnage of World War I, chemotherapy arose as a conscious harnessing of chemical warfare and its cytotoxic properties. Faced with fears of a Second World War and deployment of the mustard gas that proved so deadly during World War I, the U.S. Office of Scientific Research and Development commissioned censored studies on chemical agents for allied use. Yale pharmacologists Louis Goodman and Alfred Gilman were assigned the task of elucidating the toxic and therapeutic properties of nitrogen mustard [46]. Using animal models challenged with lymphoid tumors, Goodman and Gilman observed marked regressions in tumor size that prompted further investigation into the chemical’s cytotoxic effects [47,48,49]. In collaboration with their colleague and thoracic surgeon Gustaf Lindskog, the trio admitted into their study a male patient with advanced radiation-resistant lymphosarcoma. The event marked the first documented case of intravenous chemotherapy treatment, and was maintained in secrecy until its publication nearly four years later in 1946 [50]. While the patient ultimately succumbed to treatment, the trial imparted two key observations on nitrogen mustard. First, delivery of the compound induced immediate tumor regression and short-term remission. Second, the absence of a durable response was exacerbated by chemoresistance and severe bone marrow suppression, important features that continue to plague modern-day chemotherapies [51].

Goodman and Gilman’s early work on nitrogen mustard fed the frenzy to screen the compound, its derivatives, and alternative chemical agents in animal models of cancer as well as in patients. The latter half of the 20th century saw the emergence of several new chemotherapies whose development was largely supported by WWII-related initiatives [52]. Nutritional studies led by Sidney Farber (1903–1973) supported the development of folic acid antagonists that produced unprecedented but temporary remission rates in children with acute lymphoblastic leukemia [53]. Among the tested compounds, a 4-amino derivative of folate called aminopterin was used for the development of a less toxic analogue now known as methotrexate (formerly amethopterin). In subsequent studies pioneered by Jane Wright (1919–2013), methotrexate was shown to induce durable remission in adults with incurable neoplastic disease. Notably, of ten patients with metastatic breast carcinoma, three responded favorably to treatment, and 54 of the 93 enrolled patients showed some measure of improvement [54]. Wright’s study was the first to both test and prove a positive association between methotrexate and remission in solid malignancies, thus paving a path forward for its continued use in modern oncology.

Several other chemotherapies seemingly materialized overnight and were rapidly approved by the U.S. Food and Drug Administration. Among these, the purine analogue 6-mercaptopurine synthesized by Gertrude Elion (1918–1999) was used against childhood leukemia and earned Elion the 1988 Nobel Prize [55,56]. This was followed by the development and application of cyclophosphamide, a mustard-based derivative and alkylating agent found to disrupt DNA synthesis [57]. Charles Heidelberger (1920–1983) later developed the first major chemotherapy targeted to non-hematological malignancies. The fluorine pyrimidine 5-fluorouracil demonstrated tumor-inhibitory effects across a wide spectrum of solid tumors and was quickly adopted for use in patients [58]. However, it was not until after the 1960s that chemotherapy became accepted as a beneficial treatment for metastatic malignant disease. Chinese-American oncologist Min Chiu Li (1919–1980) forged a new paradigm for chemotherapy after he noticed declining levels of circulating human chorionic gonadotropin (hCG) in patients receiving methotrexate. Li hypothesized that some tumors actively secrete hCG, and chemotherapy-induced tumor shrinkage could be monitored through urine hCG levels. Li tested his controversial theory on three patients with metastatic choriocarcinoma by using hCG to determine the frequency, dosage, and duration of methotrexate treatment [59]. In opposition to the standard of care, Li insisted that patients with high residual hCG levels should continue to receive chemotherapy, despite demonstrating complete remission in metastatic disease. Although the National Cancer Institute criticized Li’s actions as unprecedented and unnecessary, Li was later exonerated by reports of relapse in patients whose chemotherapy was suspended [52]. Li’s intuitive use of hCG as the first circulating tumor biomarker was followed by yet another original and impactful discovery; combination treatment with multiple chemotherapies was effective in cases of metastatic malignancy [60].

To combat waning support throughout the next decade, medical oncologists curated new applications for chemotherapy. Combination chemotherapy emerged in the late 1960s as a viable treatment for advanced breast cancer [61], with a cyclical delivery regimen consisting of cyclophosphamide, methotrexate, and fluorouracil (CMF) inducing response rates exceeding 50% [62]. The advent of adjuvant chemotherapy arrived with the publication of several prominent studies spearheaded by Gianni Bonadonna and Bernard Fisher. Bonadonna and Fisher, respectively, investigated the administration of CMF and L-phenylalanine mustard to breast cancer patients following mastectomy and demonstrated overwhelmingly favorable results [63,64]. Breast-conserving surgeries rose to popularity as an increasing number of studies showed improved remission rates associated with lumpectomy followed by adjuvant combination chemotherapy [65,66]. Following the success of adjuvant chemotherapy, medical oncologists expanded their practice to include other modalities of chemotherapy delivery. Among these, neoadjuvant chemotherapy increased remission rates by inducing tumor shrinkage prior to surgical resection [67,68]. 

Spurred by renewed public interest on the “war on cancer” and the passage of the National Cancer Act in 1971, new classes of chemotherapies were rapidly introduced and approved for patient use. The application of improved screening models, including the use of human xenografts in nude mice, was critical to shaping their success [69,70]. During this period, a class of DNA-intercalating agents isolated from *Streptomyces* bacteria gained international fame for exerting broad anti-tumor effects across several cancers [71,72]. Known as the anthracyclines, the family includes the widely used doxorubicin and is still administered in up to 30% of breast cancer cases [73]. Moreover, neoadjuvant delivery of anthracyclines in certain cases of aggressive breast cancer has been shown to improve patient response [74]. 

In addition to the anthracyclines, the taxanes encompass another class of anti-tumor compounds isolated from natural sources that demonstrated clinical success. Originally derived from *Taxus brevifolia* (the Pacific yew tree), paclitaxel and docetaxel were shown to induce cell cycle arrest by stabilizing tubulin and preventing mitotic division [75,76,77]. The taxanes were heralded for their novel mechanism of action and ability to induce impressive remission rates throughout the 1990s. Presently, the use of paclitaxel and docetaxel remains a popular choice by medical oncologists for the treatment of breast cancer and other solid malignancies [78].

### 2.3. Targeted Therapies Take Hold

Support for a hormonal basis for cancer burgeoned half a century after Beatson’s original studies on applying oophorectomy to treat breast cancer. Critical work led by Charles Huggins (1901–1997) at the University of Chicago demonstrated regression of metastatic prostate carcinoma following removal of the testes [79,80]. Huggins conclusively showed that cancer cells were dependent on endogenous hormone production, and removal of such chemical signals was effective in halting advanced metastatic disease [81,82]. Huggins was later awarded the 1966 Nobel Prize for his work in fortifying the endocrine axis in cancer etiology, which birthed a new paradigm for cancer treatment. A new wave of endocrine therapies began with additive delivery of steroid compounds, such as estrogen [83], androgen [84], and corticosteroids [85] to patients with metastatic breast cancer, and evolved with growing knowledge on the mechanisms that regulate estrogen signaling [86]. 

In 1966, an accidental discovery made by scientists at the England-based ICI Pharmaceuticals led to the development of the first anti-estrogen compound that later became the gold standard in breast cancer treatment. In an effort to screen potential anti-fertility compounds for contraceptive use, a team led by Arthur Wapole identified a potent anti-estrogen previously synthesized and supplied by organic chemist Dora Richardson. Compound ICI 46,474 later proved ineffective as a contraceptive, but at Wapole’s insistence, was carried into preclinical and phase II clinical trials for palliative treatment of advanced breast cancer [87]. Further testing of the compound in the U.S. revealed that it acted as a selective estrogen receptor modulator (SERM) and blocked 3[H]-estradiol-mediated ER activation. In a novel finding, ICI 46,474, now reinvented under the name tamoxifen, was shown to block both the growth of rat mammary carcinoma, as well as chemical induction of carcinogenesis [88]. These results prompted application of the hormonal therapy in clinical trials, where it was shown to induce remission in women with early and node-positive breast cancer [89,90,91]. Currently, long-term delivery of tamoxifen (i.e., a five- or ten-year course) is commonly used to treat pre- and postmenopausal women with ER+ breast cancer [92].

Subsequent years saw the development of other SERMs, in addition to a new class of estrogen-modulating compounds for the treatment of advanced breast carcinoma. The discovery that testosterone could be converted to estrogen through a process called aromatization laid the foundation for further research into aromatase inhibitors [93,94,95,96]. Aminoglutithemide was among the first of such compounds to be adopted as a second-line therapy for breast cancer; however, the drug failed to match the efficacy and safety profile of tamoxifen. Despite achieving up to 90% aromatase inhibition, aminoglutithemide and subsequent second-generation aromatase inhibitors struggled to demonstrate robust suppression of estrogen levels [97]. To address these concerns, a third-generation of more potent aromatase inhibitors were developed. Compounds within this group, including anastrozole, letrozole and exemestane, demonstrated up to 98% aromatase inhibition and estrogen suppression [98,99,100,101]. Subsequent phase III clinical trials comparing these compounds to tamoxifen revealed superior remission rates in both early and metastatic breast cancers, thus securing their spot amidst modern-era therapies [102,103,104].

The introduction of DNA and genome sequencing technologies during the late 1960s led to the identification of numerous genetic targets for therapeutic intervention. Identification of the first oncogene in 1970 bolstered the search for additional oncogenes and proto-oncogenes regulating growth factor signaling and carcinogenesis [105,106]. These events spurred the development of targeted therapies to suppress signaling at the transcriptional and translational levels. The expansion of hybridoma technologies in the early 1970s enabled production of monoclonal antibodies that targeted various mitogens, receptor tyrosine kinases, and transcription factors frequently over-activated in human cancers [107,108,109]. When delivered alongside chemotherapy, targeted antibodies improved remission rates beyond those experienced with chemotherapy alone [110]. Such antibodies were most frequently generated against cellular transmembrane receptors functioning as signal transducer elements, and thus were commonly referred to as signal transducer inhibitors (STIs). Among the most popular STIs are trastuzumab and pertuzumab, recombinant antibodies targeted to Human Epidermal Growth Factor Receptor 2 (HER2 or ErBB2) [111,112,113]. Trastuzumab, commercially known as Herceptin, and its close relative pertuzumab may be administered alongside chemotherapies (typically with paclitaxel or anastrozole) to patients with HER2-amplified breast cancer [114,115]. Other prominent STIs that emerged in breast cancer treatment include everolimus, a second-generation inhibitor of the Mammalian Target of Rapamycin (mTOR) [116,117], and inhibitors of Cyclin-Dependent Kinases (CDKs) [118]. These compounds suppress mitogenic signaling pathways often required for supporting solid tumor growth.

### 2.4. Immunotherapy: A Third Pillar Is Erected

The 2018 Nobel Prize in Physiology and Medicine was awarded to James Allison and Tasuko Honjo, two immunologists whose original discoveries significantly impacted modern cancer treatment. Thus, it might seem startling that a glimpse into their nascent careers would reveal pervasive skepticism surrounding the notion of immune modulation to combat cancer [119]. Indeed, the journey toward understanding the role of immunity in cancer and asserting a path for therapeutic targeting has long been a tumultuous one. The first immune-based therapy to treat neoplastic disease was introduced by surgeon William Coley (1862–1936) during the late 19th century. After researching cases where concomitant infection in cancer patients improved their conditions, Coley conceived the idea that infection-induced immunity could effectively eliminate cancer cells. After cases of fatal trial and error, Coley discovered that injecting patients with a mixture of heat-inactivated Gram-positive and Gram-negative bacteria induced complete disease remission [120,121]. Then, viewed as haphazard, Coley’s actions were largely condemned by the medical establishment and the use of “Coley’s fluid” was laid to rest in subsequent decades [122]. 

The establishment of the New York Cancer Research Institute in 1953 by Coley’s daughter, Helen Coley Nauts, supported the re-emergence of harnessing the immune system to treat cancer [123]. These efforts were bolstered by a series of seminal discoveries that elucidated the function of T cells [124], dendritic cells [125], and natural killer (NK) cells [126]. Subsequent years saw the introduction of immune-based therapies to treat hematological malignancies, including bone marrow transplantation and recombinant interferon delivery [127]. Further evidence to support T cell-mediated tumor immunity came in 1998 and revealed an interferon gamma (IFNγ)-dependent mechanism for inducing tumor immune surveillance [128]. Subsequent work showed that cooperative signaling between lymphocytes (T, NK, and B) and IFNγ mediates tumor cell pruning; however, gradual selection pressure exerted through this process leads to the enrichment of tumor cells demonstrating reduced immunogenicity, thus supporting a direct route toward tumor immune evasion [129]. 

The immuno-oncology field slowly garnered interest after multiple independent studies identified the Cytotoxic T-Lymphocyte Antigen 4 (CTLA-4) as a negative regulator of T cell activation and proliferation [130,131]. In the same year, James Allison’s team at UC Berkeley demonstrated that CTLA-4 activation was one of two opposing signals (the other being CD28 stimulation) integrated by T cells that determine an anti-tumor response [132]. Allison’s group provided definitive evidence to support CTLA-4′s role as a critical immune checkpoint that could be inhibited to enhance anti-tumor immunity [133]. Despite igniting interest into this regulatory mechanism, it was not until a decade later that the first CTLA-4 inhibitor, ipilimumab, entered phase III clinical trials. Ipilimumab was shown to improve survival rates for patients with stage IV melanoma, thus fast tracking its approval by the FDA for treatment of melanoma in 2011 [134].

In the years leading up to the discovery of CTLA-4 as a critical immune regulator, an independent team at Kyoto University had taken similar steps to characterize a related and then-unknown immune checkpoint. Led by Tasuko Honjo, the group was credited for isolating Programmed Cell Death 1 (PD-1) on murine T cells and was among the first groups to identify its involvement in autoimmune function and negative regulation of T cell activation [135,136]. Subsequent studies identified an antigen that directly engaged with PD-1 to elicit the previously observed immuno-inhibitory effects [137,138,139]. Initially called B7-H1, the antigen was later renamed Programmed Cell Death Ligand 1 (PD-L1) and revealed to be highly expressed on stromal cells adjacent to T cells within the tumor microenvironment [140]. Further research identified a PD-L1 isoform (PD-L2) and additional binding partners, including those that were previously shown to engage with CTLA-4 [141,142,143]. Accumulating evidence to support disruption of PD-1/PD-L1 as a cancer therapeutic led to the development of multiple PD-1/PD-L1 checkpoint inhibitors [144]. The most prominent of these include pembrolizumab, the first anti-PD-1 therapy to receive FDA approval, in addition to nivolumab (anti-PD-1), atezolizumab (anti-PD-L1), and avelumab (anti-PD-L1) [145]. Clinical trials comparing anti-PD-1/PD-L1 therapies to ipilimumab showed improved survival outcomes and reduced toxicity associated with the former [146].

Currently, over a dozen immunotherapies have been approved for use in various cancers, including melanoma, hematological malignancies, and cancer of the lung, liver, bladder, colon, stomach, head and neck. Additional FDA approval has enabled their use as second- and third-line therapies in cases of multiple chemoresistance or as indicated by positive biomarker expression. While immunotherapies have achieved remarkable success in several of these cancers, overall response to these therapies in breast cancer remains modest [147]. The lackluster objective response rates observed in breast cancer can in part be attributed to fundamental differences in the immunobiology of the disease and the presence of opposing signatures between subtypes [148,149,150]. Key features dictating immune checkpoint sensitivity in melanoma and other immuno-responsive cancers (i.e., high neo-epitope load and somatic mutational burden) are relatively less abundant and predictable in many breast tumors [151,152,153,154]. Furthermore, PD-L1 expression in metastatic breast cancer is comparatively low, with up to 80% of biopsies showing less than 1% of PD-L1-expressing tumor cells [155]. Thus, therapies intended to activate or target these features remain limited in their efficacy.

Nevertheless, multiple modalities of immune-based therapies for breast cancer are currently being evaluated in multi-phase clinical trials, including adoptive T cell therapy, preventative vaccines, and immune checkpoint blockade. In addition to those already tested in clinical trials, other emerging therapies involve the use of Chimeric Antigen Receptor-engineered T (CAR-T) cells, oncolytic viruses, and exogenous interferon delivery [147]. Stemming from the immuno-stimulatory properties of “Coley’s fluid”, these immunotherapies aim to launch an anti-tumor response by averting T cell exhaustion, bolstering tumor infiltration, and priming T cell antigen presentation. As evidence of the shifting paradigm, results from the IMPassion130 clinical trial (NCT02425891) suggest promise in combination delivery of anti-PD-L1 inhibitors with existing chemotherapies, particularly in patients with triple negative breast cancer (TNBC). In a historic move, the FDA granted accelerated approval for the use of atezolizumab in combination with chemotherapy in patients with non-operable metastatic TNBC [156].

## 3. Progression toward Personalized Medicine and Closing Remarks

The 21st century has seen modern cancer therapies flourish, from the novel application of targeted and combination chemotherapies to the development and delivery of near-curative immunotherapies. Invigorating still are the vast technologies and therapies positioned on the horizon. Where just decades prior scientists struggled to elucidate the basic structure of DNA, modern-day researchers have unraveled complexities in its organization, regulation, and processing that were previously deemed inconceivable. The convergence of epigenetics, diet, lifestyle, environmental exposures, and inherited factors is now widely recognized as predicting risk and survival outcomes. The clinical manifestation of this knowledge has birthed a new paradigm of personalized medicine that prioritizes the individual patient when deciphering available treatment options.

No longer shrouded in complete mystery, tumor cells, and their respective host environments are now being illustrated in vivid detail. Real-time assessment of circulating tumor cells, DNA and various biomarkers weaves an epic tale of tumor evolution and overall disease progression. This information guides modern-day treatment and enables the precise deployment of second- and third-line therapies to combat drug-induced resistance. Further, a plethora of biomarker panels are currently in development and being tested for application in early cancer diagnosis and risk assessment. Emergent nanotechnologies have directly facilitated these events and include novel detection-based systems designed to push the limits of bioassay sensitivity.

## 4. Conclusions

Despite achieving improved remission rates over the past two decades, cancer researchers and clinicians continue to face significant hurdles in their efforts to develop preventative and curative therapies. Among these challenges, the development of therapeutic resistance continues to perplex researchers and plague patient response. Nevertheless, emergent knowledge on tumor–stromal interactions and their spatiotemporal evolution in the face of therapy will undoubtedly inform novel treatment approaches to avert adaptive resistance. Further, as researchers’ toolkits continue to expand and intersperse across disciplines, the need to assimilate and interpret a growing sea of data remains critical. Thus, an imminent priority lies in curating new platforms and pipelines for data acquisition, management, and integration into the clinic.

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
