# Peer review of "Evolution of Medical Approaches and Prominent Therapies in Breast Cancer"

_cancers, 2022, doi:10.3390/cancers14102450_

Round 1

Reviewer 1 Report

The review work "Evolution of Cancer Research, Prominent Therapies and Clinical Characterization of Breast Cancer" provides a profound and very well structured overview of the history of the breast cancer therapies and their evolution over time. The review ascends from as early as Antiquity and culminates with the description of the state-of-art approaches, such as targeted therapies, immunotherapies and personalized medicine. The language is rich and clear, and is thus very appropriate to convey the message. Surely this kind of rare and timely historical overview would be useful and interesting for a broad range of readers.

Certain points can be raised though:
- The obvious mismatch of the contents and the title - while for sure the manuscript speaks about the evolution of "Cancer research" and "Prominent therapies", the aspect of "Clinical characterization" evolution is actually nearly absent. I'd suggest to remove this from the title
- Additionally, the title repeats the word "cancer" two times making it sound clumsy
- A typo - Marie Curie's full name is spelled as Marie Sklodowska-Curie

Author Response

Reviewer 1 Comments:

The review work "Evolution of Cancer Research, Prominent Therapies and Clinical Characterization of Breast Cancer" provides a profound and very well structured overview of the history of the breast cancer therapies and their evolution over time. The review ascends from as early as Antiquity and culminates with the description of the state-of-art approaches, such as targeted therapies, immunotherapies and personalized medicine. The language is rich and clear, and is thus very appropriate to convey the message. Surely this kind of rare and timely historical overview would be useful and interesting for a broad range of readers.

Certain points can be raised though:

  1. The obvious mismatch of the contents and the title - while for sure the manuscript speaks about the evolution of "Cancer research" and "Prominent therapies", the aspect of "Clinical characterization" evolution is actually nearly absent. I'd suggest to remove this from the title.

We thank the Reviewer for pointing out the discrepancy between the title and main text and have removed the “Clinical characterization” statement from the title.

  1. Additionally, the title repeats the word "cancer" two times making it sound clumsy

We have amended the title accordingly to enhance readability.

  1. A typo - Marie Curie's full name is spelled as Marie Sklodowska-Curie

The name has been corrected accordingly.

Reviewer 2 Report

This is an interesting and instructive article summarizing in a few pages the history of breast cancer diagnosis and treatment

The manuscript is well organized and this reviewer has very few specific comments on the content and few suggestions for minor edits:

§ 1.1 line 23 B.C. is an abbreviation of before Christ, while AD (lines 42 and 48) in latin means anno domini, i.e., in the year of our Lord. Then line 35 there is also BCE (Before Common Era) , and the reader may be a little confused about when exactly that discovery was made. This reviewer is suggesting to be consistent in dating events of the very ancient times

§ 1.2 the word humorism (line 67) can't be found in a English dictionary, I understand what the authors mean with that word but is it a bit unusual?

§ 2 this reviewer has been kind of wondering why the title reads "reign and fall": was there ever a real fall?

§ 2.2 this reviewer would change "human patients" with "patients" (lines 177 and 198)

§ 2.3 lines 269-71 tamoxifen is called adjuvant chemotherapy, and induced remission in  women with early and node positive breast cancer;  this reviewer thinks it can be misleading to call tamoxifen adjuvant chemotherapy since it is hormonal therapy; also, it can't induce remission in the adjuvant setting because there is no evidence of disease in such setting

§ 2.3 lines 302-4 trastuzumab and pertuzumab administered as combination chemotherapies, but these are monoclonal antibodies

§ 2.3 lines 363-65 my reading is that anti-CTLA-4 is inferior to anti-PD/PDL-1 because of higher autoimmune toxicity associated with anti-CTLA-4 therapy. However, such sentence mixes efficacy and safety when the authors are talking about superior response rates, i.e., efficacy  

Author Response

Reviewer 2 Comments:

This is an interesting and instructive article summarizing in a few pages the history of breast cancer diagnosis and treatment. The manuscript is well organized and this reviewer has very few specific comments on the content and few suggestions for minor edits:

  • line 23 B.C. is an abbreviation of before Christ, while AD (lines 42 and 48) in latin means anno domini, i.e., in the year of our Lord. Then line 35 there is also BCE (Before Common Era) , and the reader may be a little confused about when exactly that discovery was made. This reviewer is suggesting to be consistent in dating events of the very ancient times

We thank the Reviewer for pointing out this discrepancy and have adjusted the usage of the terms to B.C.E and C.E. to remain consistent.

  • the word humorism (line 67) can't be found in a English dictionary, I understand what the authors mean with that word but is it a bit unusual?

The term humorism has been removed.